# The selection of software and database for metagenomics sequence analysis impacts the outcome of microbial profiling and pathogen detection

Ruijie Xu[1,2], Sreekumari Rajeev[3], Liliana C. M. Salvador[1,2,4]*

**1** Institute of Bioinformatics, University of Georgia, Athens, Georgia, United States of America, **2** Center for the Ecology of Infectious Diseases, University of Georgia, Athens, Georgia, United States of America, **3** Department of Biomedical and Diagnostic Sciences, College of Veterinary Medicine, University of Tennessee, Knoxville, Tennessee, United States of America, **4** Department of Infectious Diseases, College of Veterinary Medicine, University of Georgia, Athens, Georgia, United States of America

☯ These authors contributed equally to this work.
¤ Current address: School of Animal & Comparative Biomedical Sciences, University of Arizona, Tucson, Arizona, United States of America
* lilianasalvador@arizona.edu

**Data Availability Statement:** The simulated mice gut microbiome dataset was obtained from a metagenomics software benchmarking project, the Critical Assessment of Metagenome Interpretation

## Abstract

Shotgun metagenomic sequencing analysis is widely used for microbial profiling of biological specimens and pathogen detection. However, very little is known about the technical biases caused by the choice of analysis software and databases on the biological specimen. In this study, we evaluated different direct read shotgun metagenomics taxonomic profiling software to characterize the microbial compositions of simulated mice gut microbiome samples and of biological samples collected from wild rodents across multiple taxonomic levels. Using ten of the most widely used metagenomics software and four different databases, we demonstrated that obtaining an accurate species-level microbial profile using the current direct read metagenomics profiling software is still a challenging task. We also showed that the discrepancies in results when different databases and software were used could lead to significant variations in the distinct microbial taxa classified, in the characterizations of the microbial communities, and in the differentially abundant taxa identified. Differences in database contents and read profiling algorithms are the main contributors for these discrepancies. The inclusion of host genomes and of genomes of the interested taxa in the databases is important for increasing the accuracy of profiling. Our analysis also showed that software included in this study differed in their ability to detect the presence of *Leptospira*, a major zoonotic pathogen of one health importance, especially at the species level resolution. We concluded that using different databases and software combinations can result in confounding biological conclusions in microbial profiling. Our study warrants that software and database selection must be based on the purpose of the study.

(CAMI) initiative (38), available at (https://doi.org/10.4126/FRL01-006421672). The raw sequence files (FASTQ) were submitted to the NCBI Sequence Read Archive under the Bioproject accession number: PRJNA717669. The individual isolates can be accessed under the following Biosample accession numbers: SAMN18507082 - SAMN18507091. All scripts for this publication are freely available on the following Github link: https://github.com/rx32940/Metagenomics_tools.

**Funding:** The sequence analysis work was supported by the National Science Foundation under Grant No. DGE-1545433 to R.X. and startup funds to L.C.M.S. from the University of Georgia Office of Research. The sample collection, sequencing and analysis was done during S.R.'s tenure at the Ross University School of Veterinary Medicine, Saint Kitts and it was supported by internal grants from the Center for One Health and Tropical Medicine.

**Competing interests:** The authors have declared that no competing interests exist.

## Introduction

Studies analyzing the compositions of microbial communities are frequently used in diverse study fields, such as ecology [1, 2], agriculture [3, 4], human/animal health [5–7], and pharmacology [8, 9]. The advancement of Next-Generation Sequencing (NGS) technologies has provided researchers with a set of culture-independent tools that identify pathogens directly from DNA sequences [10] and have emerged as popular tools for microbial profiling and pathogen detection [6, 11].

Taxonomic profiling analysis in the metagenomics discipline uses two popular sequencing approaches: 16S rRNA amplicon sequencing and shotgun metagenomics sequencing [12, 13]. Compared to the 16S rRNA amplicon sequencing approach, which only profiles bacterial and archaeal taxa [14–16], the shotgun metagenomics sequencing approach identifies all the genetic materials within a sample [17, 18] and increases the taxonomic resolution of microbial profiles by enabling microbial classification at the species-level [19]. Most importantly, shotgun metagenomics sequencing approach has broader applications for the identification of viruses and other microorganisms with simple genomes [20].

The microbial classification of shotgun metagenomics sequencing could also be divided into two primary categories: direct read profiling and assembly-based profiling [13], where software developed under each category was developed to answer different research questions. Direct read profiling software aim to quantitively characterize the microbial communities of the collected samples (e.g. species diversity and richness) [21], distinguish the presence of disease-causing pathogens from their non-pathogenic close relatives [22], and identify new microbial organisms [23]. While assembly-based classification software mainly aim to qualitatively characterize the complete genomes of uncultivated microbial organisms [24] or understand the metabolic functions of the microbial community through gene or metabolic pathway characterization (using metagenome assembly and contig binning [25, 26]).

Currently developed direct read shotgun metagenomics sequencing-based taxonomic profiling software can be separated into two groups: the alignment-based and the alignment-free software. Alignment-based software, including the tools in the BLAST suite [27–29] were thought to have high sensitivity and have been used as the standard for taxonomic profiling [13, 30]. However, these software require a significant amount of time and computational resources to build genome alignments for the high number of sequences involved in metagenomics profiling studies [31, 32]. To overcome the limitations, multiple software have been developed using different alignment-free algorithms such as those designed based on querying and storing sequences in the format of k-mers, indices, and gene-markers. In addition, some methods have been developed to improve the accuracy and sensitivity of the existing software. Previous benchmarks on shotgun metagenomic sequencing taxonomic profiling software have evaluated the software performances using either *in silico* or synthetic datasets [33–36]. However, the performances of these software in microbial profiling, community characterization, and diagnostic applications of biological specimens have been less studied. In addition, current alignment-free software require considerable computational resources for DB building and storage. Most software applications provide prebuilt DBs for users with minimal computing resources. The contents of these prebuilt DBs differ substantially between software, and sometimes, also between the prebuilt DBs distributed by the same software. Most of these software have also provided the option for users to build their own DBs with their sequences of interest. The differences in DB contents can become the source of false negatives and false positives for taxonomical profiling [30, 37], and the impact of these misclassifications has not been specifically addressed with the use of biological specimens in the previous studies [13, 30].

In this study, we aim to address the impact of using different direct read profiling metagenomic tools in microbial profiling, downstream analysis, and pathogen detection. Specifically, we aim to 1) perform standardized evaluation (precision and recall rate) for each software included in this study by comparing the real taxonomic profiles of simulated mice gut microbiome samples with the taxonomic profiles of these samples classified by each software included in this study; 2) identify differences in taxonomic profiles produced by different software and DBs combinations using biological samples collected from tissues of wild rats; 3) verify how differences between profiles of different software/DBs could become the source of biases for the downstream diversity characterizations; and 4) compare the presence of the zoonotic pathogen *Leptospira* in microbial samples detected through taxonomic profiling with the results from traditional pathogen detection methods. We evaluated the differences between profiles classified by ten different direct read profiling software and four different DBs both using a simulated dataset of mice gut microbiome and a sample dataset from biological specimens collected from *Rattus* (*R. Rattus* and *R. norvegicus*) species.

## Materials and methods

### Simulated samples

The simulated mice gut microbiome dataset was obtained from a metagenomics software benchmarking project, the Critical Assessment of Metagenome Interpretation (CAMI) initiative [38], available at (https://doi.org/10.4126/FRL01-006421672). This dataset includes 64 simulated mice gut microbiome samples from 12 different mice with samples both simulated as Illumina (pair-end, 150bp) and PacBio reads (~3000 bps/read) using NCBI's RefSeq genomes [39]. Only the first three simulated Illumina reads samples (~5GB per sample) were used in this study to produce a standardized evaluation (precision and recall) for the profiling software and DBs included in this study (2017.12.29_11.37.26_sample_0 (Sim.0), 2017.12.29_11.37.26_sample_1 (Sim.1), and 2017.12.29_11.37.26_sample_2 (Sim.2)). Since only bacterial genomes were used to generate the simulated samples, only the bacterial taxa were used to compare between the software classified profiles and the real taxonomic profiles of the simulated samples. The comparison was performed in R. The precision rate was calculated as $\frac{\text{\# of true positives taxa}}{(\text{\# of true positives taxa} + \text{\# of false positive taxa})}$ at each taxonomic level. The recall rate was calculated as $\frac{\text{\# of true positives taxa}}{(\text{\# of true positives taxa} + \text{\# of false negative taxa})}$ at each taxonomic level.

### Biological collected samples

Tissue samples from kidney (K), spleen (S), and lung (L) were obtained from four rats of two different species, *Rattus rattus* (R28) and *Rattus norvegicus* (R22, R26, and R27). Rats were captured from the island of Saint Kitts (longitude 17.3434˚ N and latitude– 62.7559˚W) following the protocols approved by the Ross University School of Veterinary Medicine (RUSVM) IACUC (approval # 17-01-04). DNA was extracted from samples using DNeasy Blood and Tissue Kits (QIAGEN Scientific Inc., MD, USA), following the manufacturer's protocol.

### Metagenomic shotgun sequencing

DNA sample quality was assessed via analysis of the DNA purity and integrity with the agarose gel. DNA purity (OD260/OD280) and concentration were measured using the Nanodrop and Qubit 2.0. The library for metagenomic sequences was constructed with 1 mg DNA per sample. Sequencing libraries were generated using NEBNextâ UltraÔ DNA Library Prep Kit for Illumina following manufacturer's instructions. The DNA sample was fragmented (350 bp),

end-polished, A-tailed, ligated with Illumina sequencing adaptor, and amplified with the PCR technique. The PCR products were then purified for sequencing. Before sequencing, samples were clustered on a cBot Cluster Generation System, then sequenced on an Illumina HiSeq platform for paired end reads.

## Data pre-processing

Sequencing adapters, low-quality reads, and host DNA reads within the sequenced samples were removed using the software KneadData (https://github.com/biobakery/kneaddata) with the default Trimmomatic [40] (version 0.33) settings (SLIDINGWINDOW:4:20 MINLEN:50) and the "—very-sensitive" Bowtie [41] (version 2.3) option. The hosts' reference genomes, which were used to separate host reads from the microbial reads, were downloaded from the NCBI's RefSeq library (*R. norvegicus*: GCF_015227675.2_mRatBN7.2; *R. Rattus*: GCF_011064425.1_RRattus_CSIRO_v1).

The human reference genome was also included in the pre-processing step for host genome filtrations to remove any potential technical contaminations within the sequenced samples (Human: GCA_000001405.28_GRCh38.p13). Quality and statistics of the sequences before and after filtering were assessed using FastQC (http://www.bioinformatics.babraham.ac.uk/projects/fastqc).

## Taxonomic profiling

Nine profiles obtained using ten different software (BLASTN [28], DIAMOND [42], MEGAN [43], Kraken2 [44], Bracken [45], Centrifuge [46], CLARK [47], CLARK-s [48], Metaphlan3 [49], and Kaiju [50]) were evaluated in this study. All software were used with the default settings according to the instruction manuals provided by the developers. One of the profiles was obtained using a combination of two software (DIAMOND and MEGAN). Software were chosen due to their frequent use in taxonomic profiling, but also to represent a wide range of database classification methods, including DNA to DNA mapping (DNA-DNA) (BLASTN, Kraken2, Bracken, CLARK, CLARK-s, and Centrifuge), DNA to markers mapping (DNA-Marker) (Metaphlan3) or DNA to amino acid mapping (DNA-protein) (Diamond+Megan and Kaiju), and mapping algorithms, including alignment-based or alignment-free (based on K-mer, Bayesian probability distributions, FM-index, and taxa specific markers) algorithms (Table 1). For a consistent comparison between the profiles provided by each software, an additional step was needed for some of the tools. BLASTN took the top one matching alignment for each read as its taxonomic assignment and the individual read assignments were further grouped together following a naïve lowest common ancestor algorithm (LCA) using MEGAN's independent blast2lca tool. DIAMOND + MEGAN (Diamond+Megan) combination was performed following a suggested protocol for profiling and binning short-read data recommended by the developer [51]. On the other hand, Bracken's profiles re-assessed the read assignments of Kraken2's profiling output using its standard DB, and CLARK-s's profiles improved the sensitivity of CLARK with the use of a spaced k-mers DB built on top of a CLARK DB. Detailed script for running each software is available in the GitHub repository: https://github.com/rx32940/Metagenomics_tools.

To test the effect of using different DBs on the taxonomic profiles of the same set of biological samples with the use of the same software, we used four different Kraken2 DBs (standard, minikraken, maxikraken, and customized). Three of the four DBs (standard, minikraken, and maxikraken) were directly obtained from publicly available resources, and the customized DB was created using Kraken2's custom DB building feature. The standard Kraken2 DB (Kraken2_std) is the default Kraken2 DB built with the complete bacterial, archaeal and the viral

**Table 1. Taxonomic profiling software and their corresponding DBs.** *Pre-built: the database was pre-built by the software developer and was distributed with the software release. **Downloaded: the database was built previously with the contribution of the science community and distributed online.

| Classification Methods (See Method) | Software | Version | Algorithm | Alignment based | Databases Used for Profiling (size of the DB) | Database Building Resources (Number of CPUs, memory usage, building time) | Software Runtime Resources (Number of CPUs, memory usage) |
|---|---|---|---|---|---|---|---|
| DNA-DNA | Blastn | v 2.12.0 | Alignment | ✓ | nt (172 GB) | Pre-built* | 12 threads, 1.75 GB |
| | Kraken2 | v. 2.1.2 | K-mer | ✗ | miniKrakenV2 (8 GB) | Pre-built* | 12 thread, 7.46 GB |
| | | | | | standard (53 GB) | Downloaded** | 12 thread, 50.54 GB |
| | | | | | maxikraken2 (150GB) | Downloaded** | 12 threads, 140.23 GB |
| | | | | | custom (60 GB) | 12 threads, 60.28 GB, ~26 hrs | 12 threads, 59.36 GB |
| | Bracken | v 2.6.1 | Bayesion proability distribution | ✗ | standard (2.7 GB) | Downloaded** | 12 thread, 0 MB |
| | CLARK | v.1.2.6.1 | K-mer | ✗ | bacteria (archaea) viruses human (168 GB) | 12 threads, 404.48 GB, ~43 hr | 12 threads, 136.83 GB |
| | CLARK-s | v.1.2.6.2 | K-mer | ✗ | bacteria (archaea) + viruses | 12 threads, (261.15 + 30.59) GB, ~37 hr | 12 threads, (289.81 + 72.03) GB |
| | Centrifuge | v. 1.0.4 | FM-index | ✗ | h+p+v+c (33 GB) | pre-built* | 12 threads, 30.62 GB |
| DNA-Marker | Metaphlan3 | v. 3.0.13 | Taxa specific markers | ✗ | mpa_v30_CHOCOPhlAn_201901_marker (2.8 GB) | pre-built* | 12 threads, 3.10 GB |
| DNA-Protein | Diamond + Megan | v. 2.0.15 | Alignment + FM-index + Binning | ✓ | nr (218 GB) | 12 threads, 7.98 GB, ~2 hr | 12 threads,9 GB |
| | Kaiju | v.1.8.2 | FM-index | ✗ | Refseq (234 GB) | 12 threads, 115.75GB, ~5 hr | 12 threads, 60.27 GB |

genomes deposited in NCBI's RefSeq libraries, along with a human genome (GRCh38) and a collection of known vectors (UniVec_Core). The standard DB used in this study was directly downloaded from the online resource, which was built, updated, and published regularly by the Benlangmead lab (https://benlangmead.github.io/aws-indexes/k2). The minikraken DB (Kraken2_mini) was directly downloaded from the main page of Kraken2. This DB was built with all the genomes included in the standard DB, but downsampled using a hash function to decrease the memory requirements for users with low computational resources. The maxikraken DB (Kraken2_max) was obtained from the Loman lab website (https://lomanlab.github.io/mockcommunity/mc_databases.html), where the DB was built with all the complete genomes included in the standard DB, plus the RefSeq's fungal and protozoan libraries and the draft or incomplete genomes deposited in the RefSeq library. Finally, the customized Kraken2 DB (Kraken2_cus) was built using all the genomes included in the standard DB, with the addition of the two *Rattus* reference genomes representing the hosts where the biological samples in this study were collected from (*R. norvegicus*: GCF_015227675.2_mRatBN7.2; *R. Rattus*: GCF_011064425.1_RRattus_CSIRO_v1).

Each of the other software used their own default standard DB. BLASTN and DIAMOND's DBs were built with NCBI's non-redundant nucleotide and protein DBs (nt/nr), respectively. Centrifuge, Kraken2, Bracken, and CLARK's DBs were built from the complete genomes in RefSeq's bacterial, archaeal, and viral libraries along with the human genome. Kaiju and CLARK-s's DBs were built with all the RefSeq prokaryotic libraries without the human genome because Kaiju's standard RefSeq DB building option does not include a human genome (kaiju-makedb -s refseq) and CLARK-s' DB building protocol was performed separately for different RefSeq libraries to avoid the technical limitation of the software. CLARK-s' DB was required to be built on top of a CLARK's DB of the same compositions, but when the

CLARK-s' DB was built with all complete genomes in RefSeq's bacterial, archaeal, viral, and human libraires, the building was suspended by the software with the error message "the number of targets exceeds the limit (16383)". This limitation was reported to CLARK-s's developer, but it has not been resolved by the time this manuscript was drafted. We bypassed this limitation by building the DB with each RefSeq's microbial libraires separately and combining the classifications using each DBs at the end of the analysis. Metaphlan3 uses a customized microbial marker DB provided by the developer. If the software has the pre-built DBs, these were downloaded directly from the software' homepage (BLASTN, Centrifuge, and Metaphlan3). Otherwise, DBs were built based on the standard instructions provided by the software' manual (CLARK, CLARK-s, Diamond, and Kaiju). The DB setup and loading process is a time-consuming step, especially for the alignment-free software (Ye *et al*., 2019). To avoid loading the DBs for every new sample, the profiling analyses of all samples were performed sequentially in a loop syntax. More information about DB building is available in Table 1.

## Analysis to determine the expected number of reads required to characterize microbial samples

The expected number of reads required to characterize the complete microbial community for the profiling result of each software was estimated using the "rarefy" function in the R package "vegan" [52] where sequenced reads in each sample were repeatedly subsampled with replacements at different read depth to estimate the number of unique microbial species can be identified at different read depth within each sample.

## Comparison of distinct species taxa identified from the different profiles

Distinct species taxa identified from all profiles of the rat tissue samples were compared in a pairwise fashion, where we defined a comparative metric, relative precision rate, to describe the differences and similarities between the distinct microbial taxa identified between two profiles classified using two different software/DBs included in this study. The relative precision rate is defined as the percentage of intersection in taxa identified from two different profiles included in a comparison (A *vs*. B) relative to the total number of microbial taxa identified by the profile A within this comparison ($\frac{|A\cap B|}{|A|}$). The relative precision analysis was performed using a custom R script.

We also assessed the between profiles relationship of different software/DBs using the Bray-Curtis (BC) indices [53], where we aggregated the number of reads classified under each microbial taxon identified from all the rat samples together to obtain a single taxonomic profile for each software/DB. The relationships between these aggregated profiles were visualized with a principal coordinate analysis (PCoA) plot using the "phyloseq" package in R [54].

## Statistical analysis

Metagenomic profiles provided by each software were loaded into R for statistical analysis using the package "phyloseq" [54]. Pairwise significant difference assessments were performed using a Wilcoxon signed-rank test implemented in R's "rstatix" package [55], which is a non-parametric statistical hypothesis test used for comparing repeated measurements on a single sample. Alpha diversity, described by the Shannon [56] and Simpson indexes [57], and beta diversity [58], described by the Bray-Curtis (BC) index [53], were used to describe the microbial diversity within and between samples, respectively, and were calculated with the R package "vegan" [52]. To determine the significant differences between rat tissue samples' microbial communities, a permutational multivariate analysis of variance (PERMANOVA) test available

in the "vegan" package was conducted among samples from different tissues using profiles classified by different software/DB. The differentially abundant (DA) taxa analyzed between samples collected from two different tissues were determined by the R package "DeSeq2" [59] using the "Wald" test, and normalized reads classified under each species taxon with the "pos-counts" method. The data visualization for the metagenomics profiles was performed using the R package "ggplot2" [60]. For all statistical analysis, p-values were adjusted with the Holm-Bonferroni method [61]. Results with p-adjusted value (padj) < 0.05 were identified as significant.

## Results

### Computational resources for DB setup and microbial profiling

DB setup and building resources as well as software runtime and memory resources are presented in Table 1. Software runtimes versus the number of reads per sample are shown in Fig 1. BLASTN and Diamond+Megan, the two alignment-based software, took the longest time to profile microbial community within each sample. Their runtimes also increased exponentially with the numbers of reads within the samples. This is because NCBI's nt and nr databases used

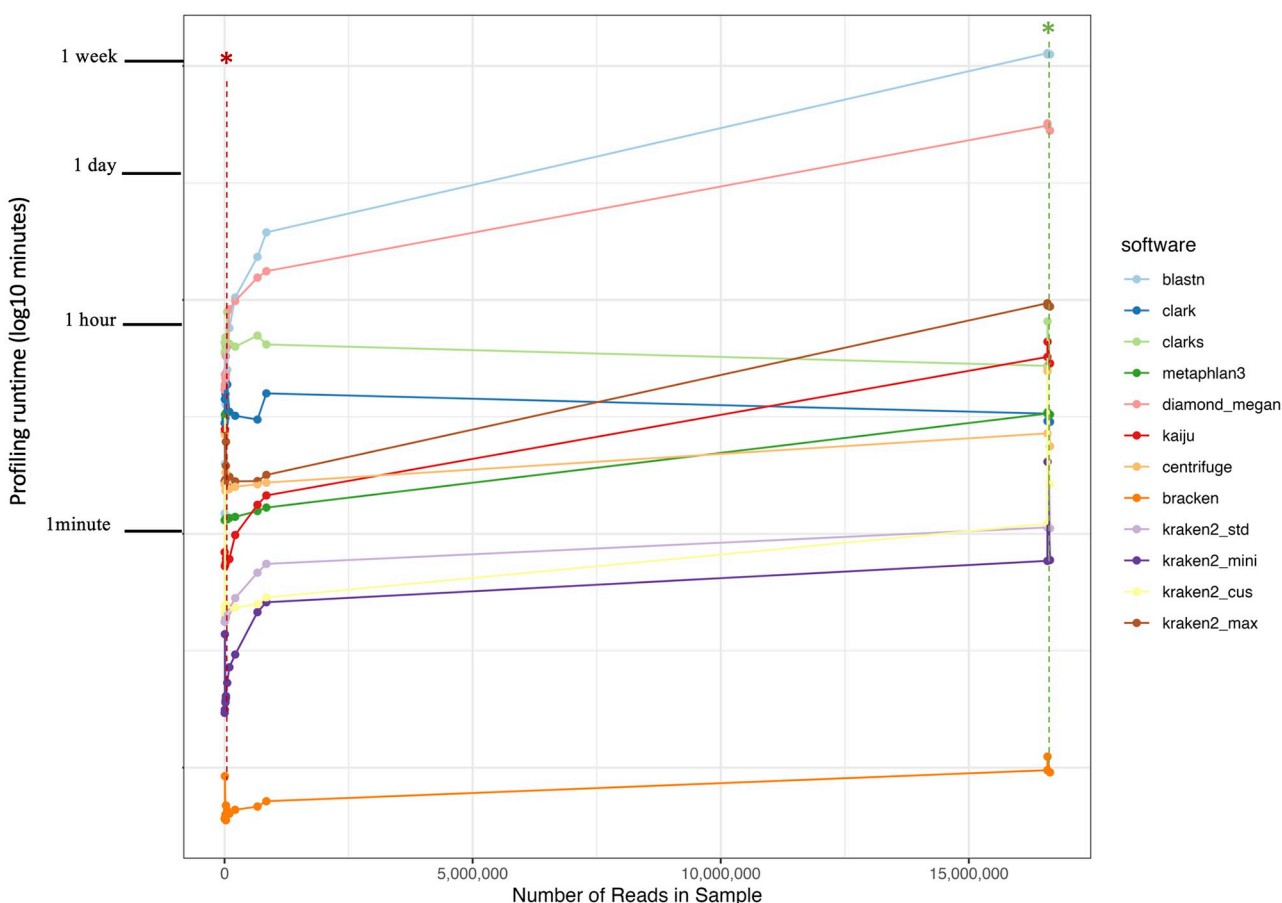

**Fig 1. Software runtimes for microbial profiling in samples of different sizes.** The runtime (in log10 minutes scale on y-axis) of each software (with different DBs) was recorded for samples of different sizes expressed in the number of reads (x-axis). Different profiles classified using different software (with different DBs) are represented by different colors. Samples labelled by the dashed line under the asterisk symbols are the first samples profiled in a sequence of profiling commands in a single task on the computing cluster (red asterisk labels the first sample profiled in the rat dataset (R22.K); green asterisk labels the first sample profiled in the simulated dataset (Sim.0)).

by BLASTN and Diamond, respectively, contain larger numbers of genomes than the NCBI's RefSeq DBs (used by most other alignment-free software included in this study) and thus require searching in a larger query space. Also, BLASTN's longer runtime was due to software's attempts to align every query read with all sequences in the DB, making this analysis computationally intensive with the current size of the nt DB. On the other hand, Diamond's algorithm, although was designed to optimize the runtime and minimize the computational requirement of BLASTX, still needs to 1) map DNA reads to the protein DB (nr DB) in a six-frame manner, which is more computationally intensive than mapping directly to a DNA DB, 2) index the DB during profiling instead of during DB building (like other alignment-free software included in the study), which trades the time for DB building with the time for profiling, and 3) align the query read to the best-matching sequence in the DB, which also contributes to the software's longer runtime.

For the alignment-free software, the profiling runtimes for CLARK and CLARK-s were the longest when classifying smaller samples. This is because both CLARK and CLARK-s load their DBs to CPU for every profiling analysis. Their runtime decreased significantly if all samples were input for profiling in a single command all together (input all samples as a list in file) to avoid loading DB repeatedly for each sample (ex. when samples were profiled sequentially using the loop syntax). In this case, only the profiling runtime of the first sample was longer due to the database loading step. The long runtime to profile the first sample also verifies in other alignment-free software. However, other alignment-free software only loads their DBs into the CPU once when profiled all samples sequentially in a single task on a computing cluster. Kraken2 runtimes are generally shorter than other alignment-free software except for when profiling using the maxikraken DB (Kraken2_max), which is around three times larger in size than the standard Kraken2 DB (Table 1). Although Bracken takes the least amount of runtime, the software does not perform the profiling analysis itself, but only corrects Kraken2 profiles' false positive read assignments.

## Benchmarking microbial profiles classified using different software/DBs

In general, the total number of reads classified are correlated with the number of distinct taxa identified in the profile, except for the profiles classified by Diamond+Megan and Metaphlan3, which have relatively fewer number of distinct taxa identified when classifying a large number of reads (S1 Fig in S1 File).

**Assess profiling software using a simulated dataset.** Before examining the differences in the taxonomic profiles of the biologically collected wild rat samples classified by different software, the performance of each software was evaluated using the taxonomic profiles of three simulated mice gut microbiome samples (Sim.0, Sim.1, and Sim.2). By comparing the software classified profiles and the real taxonomic profiles of the simulated dataset, we found that all software included in this study was able to recall all the taxa presented in the real taxonomic profiles above the species level (Fig 2a). At the species level, Kraken2_max profiles had the highest average recall rate followed by the profiles of BLASTN and Metaphlan3, while the profiles of Diamond+Megan have the lowest recall rate. Diamond+Megan and Metphlan3, when compared to the profiles classified by other software, had the highest average precision rates across all taxonomic levels with no false positive taxa reported above the family level. The profiles of these two software also have higher precision rates than the other software at the genus and species levels. Other than the Metaphlan3 and Diamond+Megan profiles, only Kraken2_-max profiles reported a high average precision rate at the lower taxonomic level, while the precision profiles of other software decreased exponentially from their phylum level classifications (Fig 2b). The lower precision rate at a higher taxonomic level (e.g. precision rate

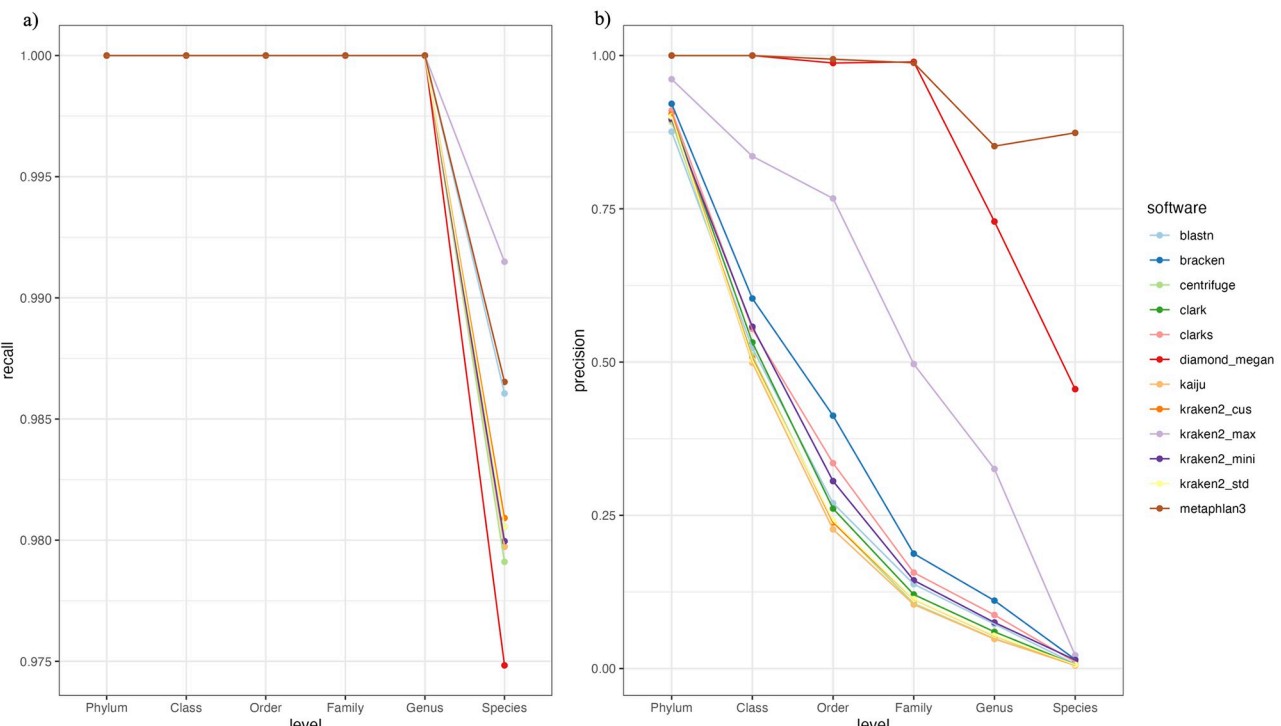

**Fig 2. The average recall (a) and precision (b) rates of the microbial profiles of the simulated samples classified by the software and DBs included in this study across 6 taxonomic levels.** Since only bacterial genomes were used to generate the three simulated samples, only taxonomic classifications under the Bacteria taxon were used to assess the performances.

of Metaphlan3 at the genus level is lower than its precision rate at the species level) was caused by the inconsistency between the taxonomy used by the real taxonomic profiles of the simulated dataset and the taxonomy used by the DBs of the profiling software. For example, Metaphlan3 has identified the presence of species taxon *Enterocloster bolteae* (taxid: 208479), which was also present in the real taxonomical profiles of the simulated dataset, however, the genus of the species taxon reported by Metaphlan3, *Enterocloster*, was not reported by the real taxonomic profiles; instead, the genus taxon, *Lachnoclostridium*, was reported as the genus taxon for this species, causing a false positive at the Metaphlan3's genus level taxonomic profile.

**Characterization of the dataset used for benchmarking.** The twelve tissue samples collected from the two rat species were used as the biological benchmarking dataset for this study, demonstrating the variations in taxonomic profiles when different software and DBs are selected. One of the significant challenges for taxonomic profiling of biological or clinical samples is the large percentage of host DNA present in the sequenced samples. Before the profiling analysis, over 99% of the sequenced reads were filtered out as host DNA. Samples with an average of 23 million reads (SD: 3,003,069) each before filtering were left with only an average of 164,610 reads (SD: 283,715) per sample (Table 2) after host DNA was removed.

After filtering the host DNA, the expected number of reads for each sample to characterize its microbial community was determined using the taxonomic profiles provided by different software and DBs (Fig 3). Each profile exhibited a different number of reads required to fully characterize the sample's microbial community. Most profiles reported by Bracken, Diamond +Megan, and Metaphlan3 reported having enough reads to fully characterize each sample's species-level microbial communities, but with a large variation in the number of unique

**Table 2. Sample information and their corresponding sequencing statistics before (Raw) and after filtering out the host DNA.**

| Sample | Tissue | Subject | Raw | Host Filtered | Filtered (%) |
|---|---|---|---|---|---|
| R22.K | Kidney | R22 | 27,171,645 | 7,012 | 99.97 |
| R22.L | Lung | R22 | 21,774,757 | 842,789 | 96.13 |
| R22.S | Spleen | R22 | 28,264,601 | 53,100 | 99.81 |
| R26.K | Kidney | R26 | 23,947,007 | 5,692 | 99.98 |
| R26.L | Lung | R26 | 22,600,630 | 214,265 | 99.05 |
| R26.S | Spleen | R26 | 18,368,987 | 1,256 | 99.99 |
| R27.K | Kidney | R27 | 24,128,657 | 18,009 | 99.93 |
| R27.L | Lung | R27 | 21,802,178 | 663,358 | 96.96 |
| R27.S | Spleen | R27 | 26,350,910 | 18,078 | 99.93 |
| R28.K | Kidney | R28 | 19,217,282 | 96,211 | 99.50 |
| R28.L | Lung | R28 | 24,887,176 | 28,084 | 99.89 |
| R28.S | Spleen | R28 | 24,564,552 | 27,472 | 99.89 |

species can be identified within each sample. The profiles of the remaining software only reported enough number of reads for microbial community characterization in some samples but not all in the dataset (e.g. R22.L sample).

## Differences in microbial profiles

The taxonomic profiles obtained in this study diverged from each other when using different software and different DBs (S1 Table), regarding the number of reads classified and the number of distinct taxa identified. With the profiles obtained from the standard DB of each software, the total number of classified reads across samples ranged from 131,460 using CLARK-s to 976,909 using Diamond+Megan, and the number of distinct taxa ranged from 18 using Metaphlan3 to 4816 using Kaiju. Regarding differences in DBs by the same software, the average number of classified reads in the taxonomic profiles of the four different Kraken2 DBs ranged from 129,061 using Kraken2_mini to 256,822 using Kraken2_max, with the identification of 1,171 to 4,589 distinct taxa.

With a wide range of eukaryotic genomes included in their DBs, the taxonomic profiles of BLASTN and Diamond+Megan classified the largest number of reads under the eukaryotic profiles with 172,276 (SD: 36.7%) and 59,553 (SD: 6.3%) reads from 1,367 and 360 distinct taxa, respectively (S2 Fig in S1 File). The eukaryotic profiles of Centrifuge, although with only one eukaryotic genome included in the DB (*Homo sapiens*), classified 27,430 (8.2%) eukaryotic reads (S2 Fig in S1 File). All the other software and DBs classified less than 10,000 (0.1% - 4%) reads under their eukaryotic profiles. The software that had *Rattus* genomes included in their DBs (BLASTN, Diamond+Megan, and Kraken2_cus) were not able to differentiate profiles from samples collected from *R. rattus* and *R. norvegicus* at their species level classification (S1 Table). BLASTN and Diamond+Megan classified over 99% of their *Rattus* reads under *R. norvegicus* for all samples, while Kraken2_cus identified approximately even numbers of reads under *R. rattus* and *R. norvegicus* species taxa across samples.

For microbial classification, all profiles, except for BLASTN (61.1%) classified over 90% of their reads under bacterial taxa with 12 (Metaphlan3) to 4,458 (Kaiju) distinct species identified (S2 Fig in S1 File). We found that profiles classified by software developed with k-mer based algorithm are more closely clustered together than the profiles classified by other software (Fig 4). Significant differences among profiles classified by software of different algorithms were validated using a permutational multivariate analysis of variance

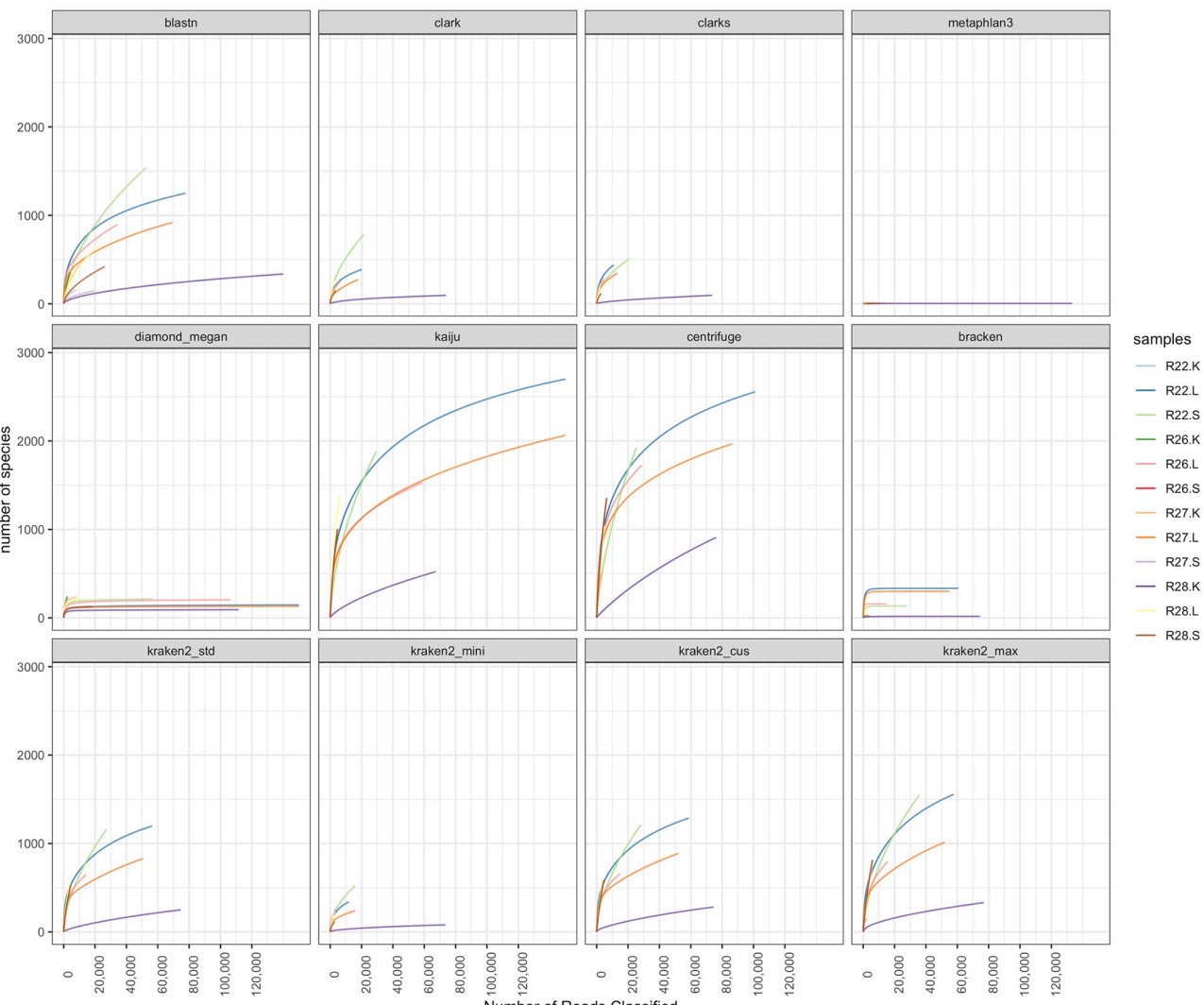

**Fig 3. The rarefaction curve depicts whether the taxonomic profiles of each software and DB (within each panel) has fully characterized the microbial community of the individual samples (represented by different colors) with the current number of reads.** The number of unique species identified (y-axis) from a set of repeatedly subsampled set of reads (x-axis) within each sample is shown in the Fig. The expected number of reads required to fully characterize the microbial profile of a sample is determined when the number of unique species identified in each sample (x-axis) no longer increases (y-axis).

(PERMANOVA) test (p < 0.05). The only profile classified by a k-mer-based software that is relatively distant from the rest of the k-mer based profiles is the profile classified using the customized DB (Kraken2_cus), which included two host genomes of the datasets in the profiling DBs. The detailed pairwise relationships between the distinct species taxa identified from the profiles of all software/DBs were reported as the "relative precision rates" (see Materials & methods) and were shown in S3 Fig in S1 File.

## Downstream analyses for microbial community characterization

**Software within-sample diversity (a-diversity).** The impact of differences in taxonomic profiles on different microbial community characterization methods is presented in S2 Table. Microbial profiles were further characterized with the two widely used microbial community

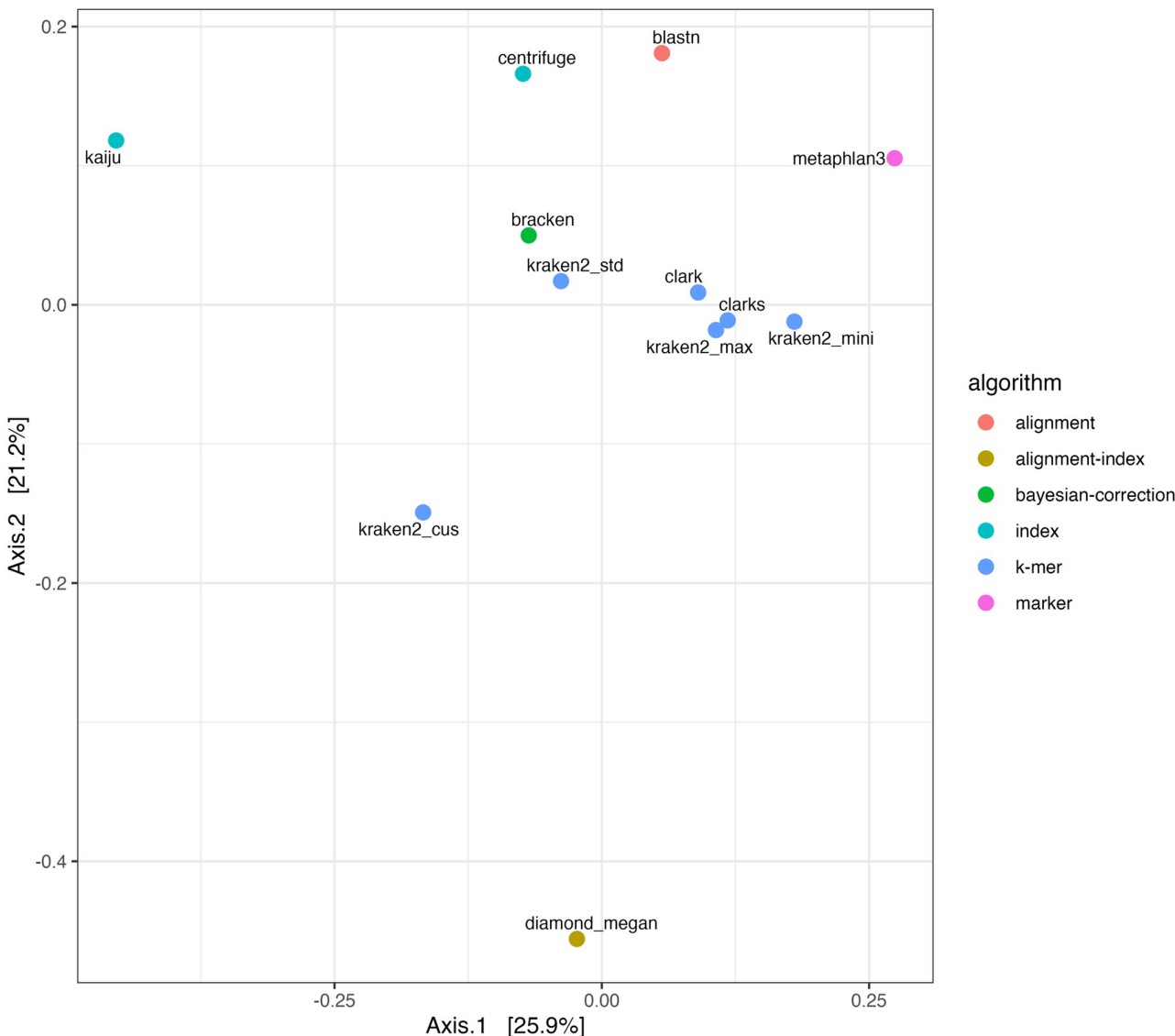

**Fig 4. PCoA plot visualizing relationships between profiles classified by different software.** The profile of each software is colored based on the type of algorithm it was developed with.

characterization indices, Shannon and Simpson indices, for community richness and evenness characterization (Fig 5). Regarding the Shannon index analysis, out of the 66 pairwise comparisons, 37 were significantly different with padj < 0.05. BLASTN showed significant differences with the two index-based software, Kaiju and Centrifuge, and the marker-based software, Metaphlan3 (Fig 5 and S2.2 Table in S2 Table). These three software were found significantly different to other software, while there is no significant difference between Kaiju and Centrifuge. In addition, the Shannon indices of the four Kraken2 DBs' profiles were primarily similar between each other, except for when compared with indices of Kraken2_mini, which were identified to be more similar to Bracken' indices (Fig 5 and S2.2 Table in S2 Table). In relation to Simpsons indices, only 5 out of 66 comparisons in the Simpson's indices were significantly different with padj < 0.05, which were less impacted by the selection of software and DBs contents (Fig 5). Most of comparisons reported significantly different in Simpson's indices were

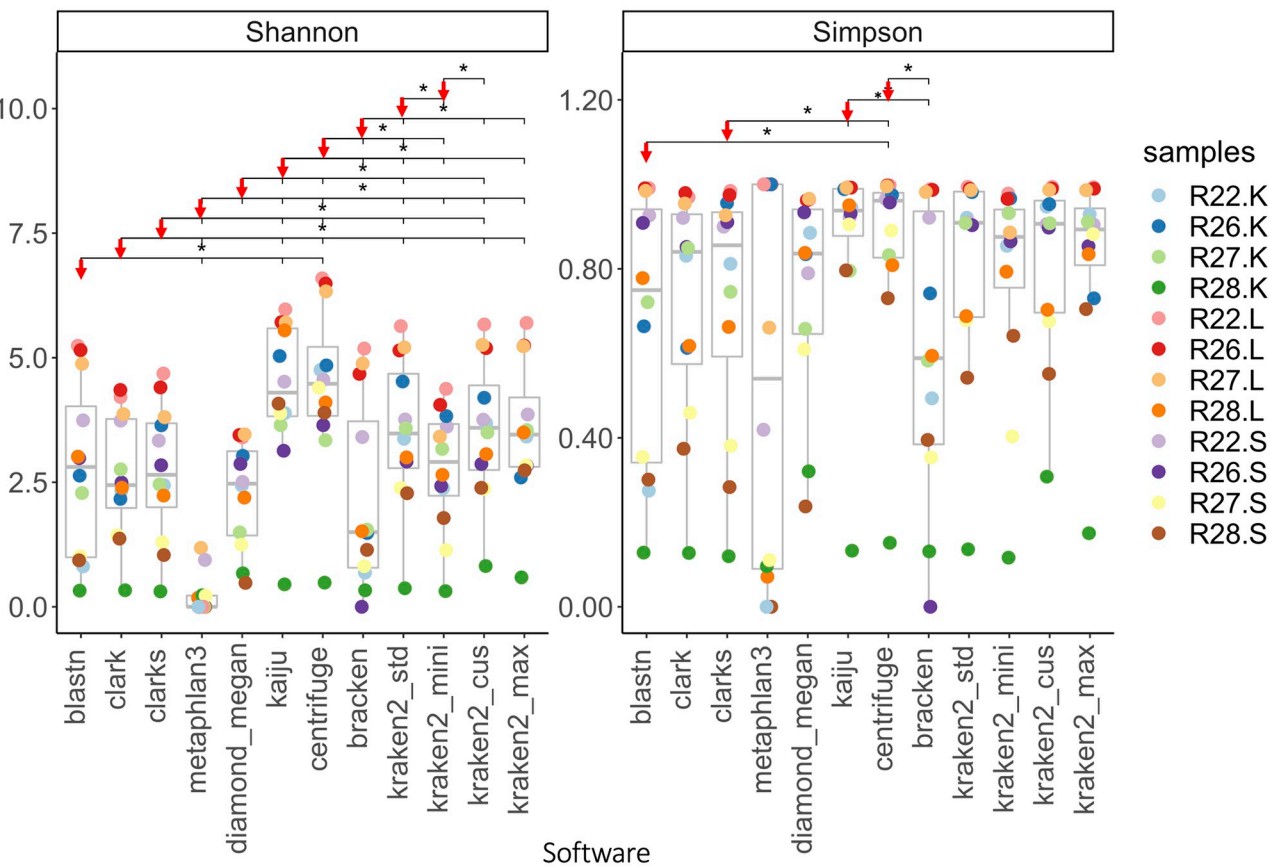

**Fig 5. The alpha indices (Shannon and Simpson) of the microbial communities characterized across samples using profiles from different software and DBs.** Indices of profiles with significant differences are shown in the fig. with padj < 0.05 labelled using significance bars above boxplots. The profile pointing by the red arrow at the beginning of each significance bar is the pivot profile of the comparisons for all comparisons shown on the same bar, which means all the profiles pointed by the black ticks on this bar are significantly different from the pivot profile the red arrow is pointing to.

either comparisons with the indices of Centrifuge, Kaiju or Bracken, or comparisons of these software's indices between each other.

**DBs between-sample diversity (b-diversity) comparisons.** In addition to the within-sample microbial community characterization, we also explored how the selection of software and DBs could impact the characterization of relationships between samples. We characterized the between sample relationship with the Bray-Curtis (BC) dissimilarity indices (S3.1 Table in S3 Table) and visualized the relationships across samples using principal coordinate analyses (PCoA) plots (Fig 6). In general, profiles of all software and DBs characterized the largest variation (axis.1) by samples collected from different rat tissues (Lung vs. Kidney and Spleen). However, the significance in differences in microbial communities among the three tissue samples evaluated using the PERMANOVA test were not consistently reported using profiles classified by different software/DBs. Here, only BLASTN, CLARK, Kaiju, Centrifuge, and Kraken_max profiles reported significant differences (p < 0.05) in the microbial profiles across the different rat tissues (Fig 6).

**Differentially abundant (DA) taxa identification.** DA taxa between samples of different tissues were identified to show taxa significantly different in abundance between the microbiomes of two tissues (S4.1 Table in S4 Table). For DA taxa identified from lung versus kidney samples at the species level, the number of DA species taxa identified using different software

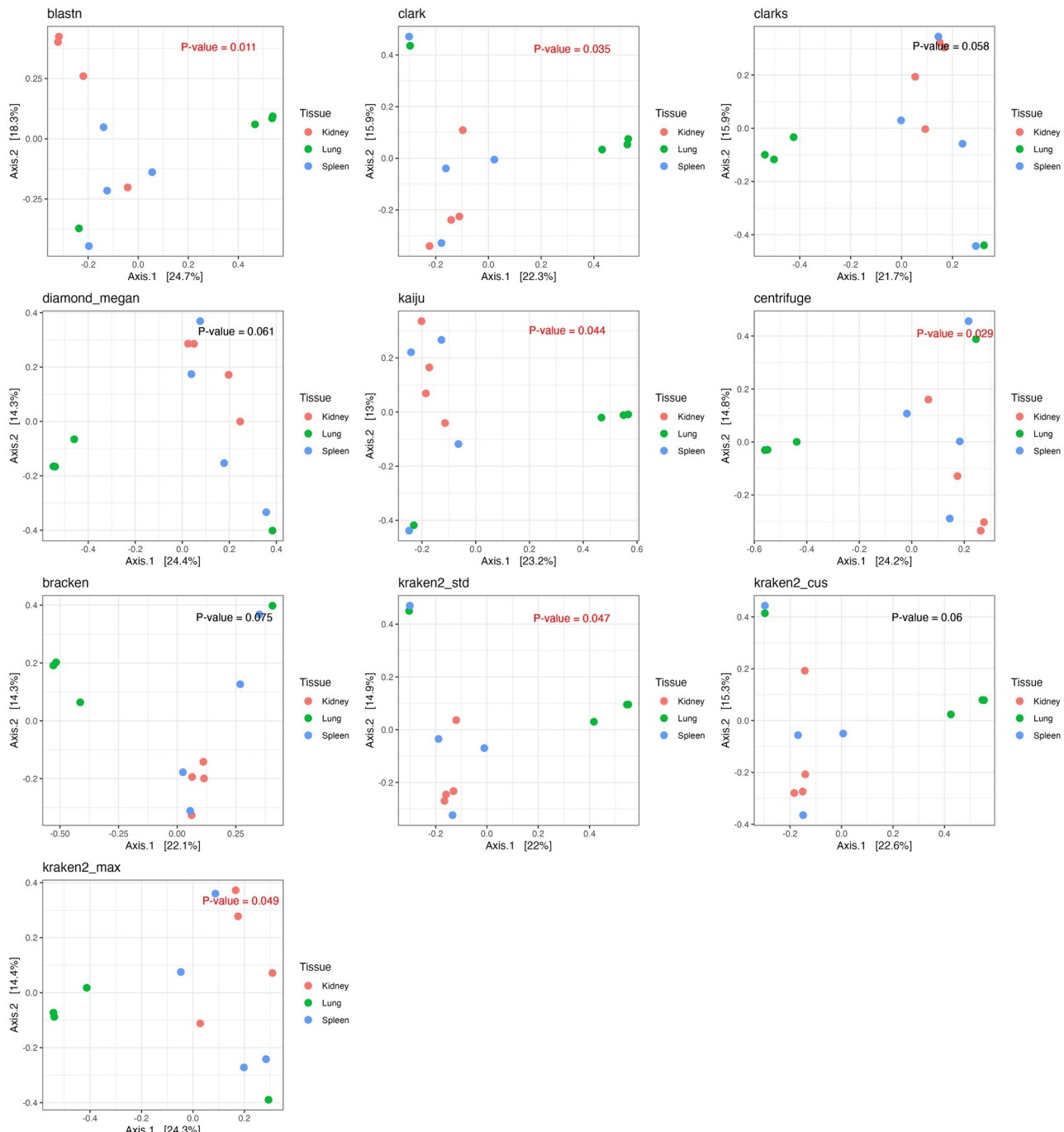

**Fig 6. Between sample diversity characterization using microbial profiles classified by different software/DBs.** Between sample diversity evaluated using the BC indices were ordinated and visualized using the PCoA plots. Significances in variations across the different rat tissue samples obtained using the profiles of different software/DBs were evaluated using the PERMANOVA test with p-values reported on the top right corner of each panel. The p-value is shown in "red" if the differences across tissues are significant.

ranged from 14 (Diamond+Megan) to 578 (Centrifuge) (Fig 7a). However, only four significantly abundant species (*Bordetella pseudohinzii*, *Leptospira interrogans*, *Leptospira borgpeterseni*, and *Mycoplasm pulmonis*) were identified by the profiles of all the software/DBs. Kaiju and Centrifuge identified the highest number of distinct DA taxa (374 and 365 taxa,

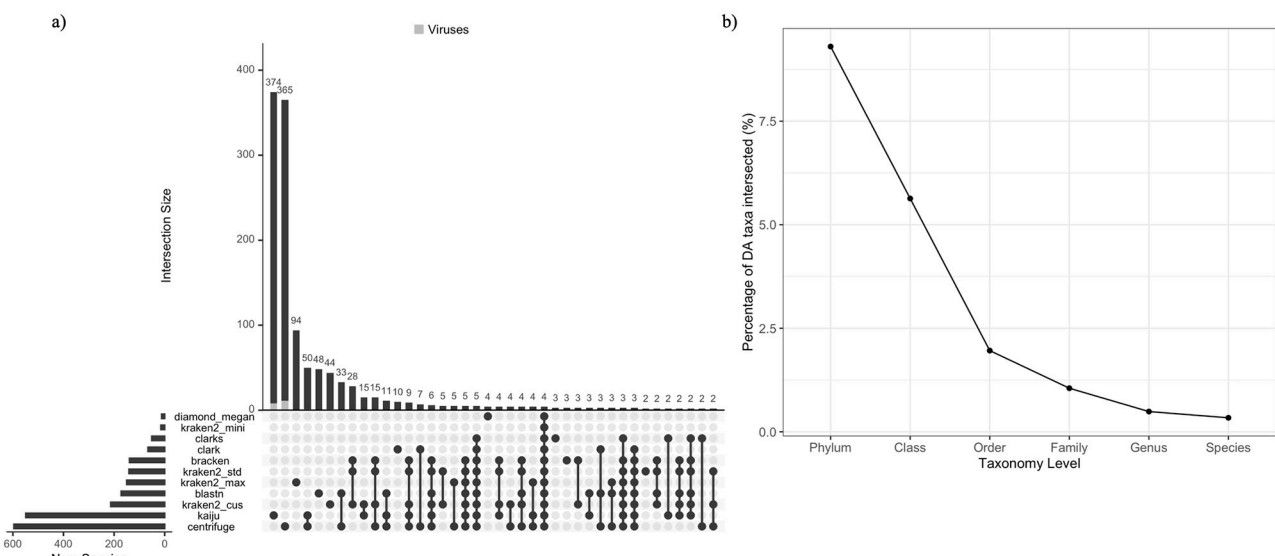

**Fig 7. UpSet Venn Diagram showing the intersection in DA taxa identified between the kidney and lung samples by different software and DB profiles. a)** Distinct intersection between the DA taxa profiles at the species level. The bottom dot plot shows the identity of the profiles included in the intersection sets, the bar plot on top shows the size of the distinctly intersected DA taxa reported in the corresponding set below, and the bar plot on the left shows the total number of DA species identified by each profile. b) Percentage of intersected DA taxa identified by all profiles at each taxonomy level over all DA taxa identified from all software and DBs' profiles.

respectively) (Fig 7a) and are the only two software that reported viral taxa as DA. In addition, these two software also reported the largest number of overlaps in the DA species identified (94 DA species) that was not reported by any other profiles in the study.

We further examined the percentage of DA taxa intersected across all profiles at different taxonomy levels (Fig 7b) and we found that the similarity between DA taxa identified by different profiles is only 9.5% at the phylum-level and decreased substantially within lower taxonomy level. The percentages of the intersection in DA taxa identified were less than 1% between the DA profiles of different software/DBs after family level (Fig 7b).

DA taxa were also identified between the microbiomes of lung and spleen (S4 Fig in S1 File, S4.2 Table in S4 Table) and between kidney and spleen samples (S5 Fig in S1 File, S4.3 Table in S4 Table). The percentage of intersection for DA taxa identified between lung and spleen samples ranges from 10.8% to 0.4% from phylum to species, with the largest decrease in DA intersection percentages observed from order-level to family-level. The percentage of intersection for DA taxa identified between kidney and spleen samples ranges from 66.7% to 0.9% from phylum to species-level.

## Pathogen detection

We first assessed the accuracy of *Leptospira* detection for each software and DBs using the simulated datasets. No *Leptospira* genome was used to generate the metagenomics sequenced reads of the three simulated samples. However, all profiles of the simulated samples classified by the software and DBs used in this study identified the presence of *Leptospira*, except for the profiles classified by Diamond+Megan and Metaphlan3. The profiles classified by BLASTN and Kraken2_cus identified as many as 11 and 10 different species of *Leptospira*, respectively, from one simulated sample. However, the false positive detection of *Leptospira sp*. in the simulated dataset was filtered out when only taxa with more than 1% in abundance from each samples' profiles was considered.

**Table 3. The identification of *Leptospira* from rat kidney samples' taxonomic profiles of different software and DBs and *Leptospira* diagnostic results using three traditional laboratory methods (PCR/DFA/culture) \*.** *Leptospira* is identified as present if at least one read is classified under *Leptospira* in the corresponding sample's profile. The number under each sample's profile is the unique number of species identified from each profile, and the number in the parenthesis is the number of pathogenic *Leptospira* species identified in the profile. *The diagnostic results were published in a previous study [62].

| software | R22.K | R26.K | R27.K | R28.K |
|---|---|---|---|---|
| BLASTN | 1(1) | | | 2(2) |
| CLARK | 1(1) | | | 4(4) |
| CLARK-s | 1(1) | | | 6(5) |
| Metaphlan3 | | | | 2(2) |
| Diamond+Megan | 2(1) | | | 7(5) |
| Kaiju | 1(1) | | 2(1) | 8(7) |
| Centrifuge | 1(1) | 1(1) | 1(1) | 4(4) |
| Bracken | 1(1) | | | 2(2) |
| Kraken2_std | 1(1) | | | 3(3) |
| Kraken2_mini | 1(1) | | | 5(5) |
| Kraken2_cus | 1(1) | | | 9(7) |
| Kraken2_max | 2(1) | 1(0) | 1(1) | 20(15) |
| **PCR/DFA/Culture\*** | +/+/+ | -/-/- | +/-/- | +/+/+ |

We further assessed the accuracy of *Leptospira* diagnostic using the rat kidney samples. *Leptospira* was previously detected in the kidney samples in the rat subjects used in this study using three traditional methods (PCR/DFA/Culture) [62]. The diagnostic results are shown in Table 3 as well as their comparison with the *Leptospira* detection using the shotgun metagenomics sequencing profiling approach. *Leptospira* was identified in the kidney samples of all three traditional methods in subject R22 and R28, and only detected in subject R27 by the PCR method.

Almost all profiles that identified *Leptospira* in the kidney samples agreed with the traditional methods in R22.K and R28.K, except for Metaphlan3, which only identified *Leptospira* in R28.K (Table 3). *Leptospira* was solely identified by PCR in sample R27.K, and was only identified by Centrifuge, Kaiju, and Kraken2_max's profiles in this study. Centrifuge and Kraken2_max also identified *Leptospira* in R26.K, which has not identified the presence of *Leptospira* by any traditional diagnostics methods. The number of pathogenic and non-pathogenic *Leptospira* species identified from profiles of different software and DBs also varies significantly (Table 3).

By filtering out taxa with less than 1% in abundance, the consistency between the *Leptospira* profiles classified by different profiles increased (S5 Table). Almost all profiles identified only one pathogenic *Leptospira* species in the kidney sample of R22 (R22.K) and 2 pathogenic *Leptospira* species in the kidney sample of R28 (R28.K). Only the BLASTN, Diamond+Megan, and Metaphaln3 profiles failed to detect the presence of *Leptospira* from the kidney of R22 and Diamond+Megan only identified one pathogenic *Leptospira* species from the kidney of R28.

## Discussion

The field of metagenomics, developed with the advancement of NGS technologies, allows scientists to build a complete and discriminatory microbial profile for samples collected from specimens of interest [63]. These metagenomic profiles can also be used to detect relevant pathogens in clinical and epidemiological investigations [64, 65] and to observe the interactions between micro-ecosystems and their changing environments [66]. Researchers achieve

this using several different taxonomic profiling software and DBs combinations. However, the selection of software and DBs can substantially impact the resulting microbial profiles of a clinical or environmental dataset [30, 34, 35]. In this study, we identified differences in the microbial profiles when different direct read taxonomic profiling software and DBs were used by analyzing simulated and biologically collected metagenomic samples. Our results show that there are differences in the classification outputs when different DBs and taxonomic profiling software are used, and these differences will substantially change the results of microbial community characterization, statistical analyses, and pathogen detection downstream of profiling.

With the use of simulated data, we evaluated each software included in this study by comparing the real taxonomic profiles of simulated samples with the taxonomic profiles of these samples classified by each software. We determined that every software was able to recall all the taxa presented above the species level, however, they differed substantially on their average precision rates across all taxonomic levels. With the use of rat samples, our study demonstrated that one of the most important challenges in taxonomic profiling analyses is the presence of overwhelmingly large amount of host DNA contamination in samples. Previous studies have demonstrated that increase in host DNA percentage in a shotgun metagenomics sequenced sample will generally decrease the sensitivity of taxonomic profiling [67] and will also lead to higher rates of misclassifications [30]. In response to this challenge, many studies have aimed to reduce the amount of host DNA during library preparation [68–70]. However, all these studies were designed to reduce host DNA contaminations for specific types of biological samples, and there is no generally effective approach for host DNA depletion before shotgun metagenomics sequencing [71]. In our study, we showed that host DNA contamination can overtake more than 99% of the reads sequenced in a tissue sample with a standard shotgun metagenomics sequencing library preparation protocol, leaving less than 1% of the reads for microbial taxonomic profiling analysis. These host DNA contaminations in metagenomics sequenced samples will not only impact the accuracy of quantitative characterization for the microbial communities, but also will prevent the potential of performing qualitative analyses using shotgun metagenomics sequenced data. For example, to perform functional analysis using the rat tissue samples collected in this study requires the assembling and binning of the sequenced reads beforehand, however, only 4 out of 12 tissue samples were able to obtain contig bins after metagenome assembly+binning, with less than 2 bins clustered from each sample (data not shown). Therefore, host DNA depletion before sequencing is a necessary step for designing a taxonomic profiling study for environmental or clinical samples.

In addition to host DNA depletion before sequencing, filtering host reads before profiling was also proposed as an essential step to reduce the impact of host contaminations [13, 30]. In this study, we found that a large percentage of reads were still classified under the host taxa after host reads were filtered. This could be the result of sequenced host DNAs having larger genetic divergence with the reference genomes used for filtering or due to the misclassification of reads from organisms that are genetically close to the host genomes. Although not being the focus of most microbial profiling studies, the presence of host reads in the biological samples could be used as a clear metric to evaluate the performance of taxonomic profiling software. Shotgun metagenomics sequencing approach was often known for its advancement in profiling microbial community at the species-level; however, with the profiling of host reads collected from two different rat species in this study, none of the taxonomic profiling software has successfully differentiated samples collected from the two different rat species. The incapability of species-level classifications using direct read metagenomics profiling has also been validated by the lower recall and precision rate obtained from the classified profiles of the simulated samples. This weakness in species level classification could be due to the divergence between the sequenced DNA and the reference genomes of the taxon included in the DBs, or

due to the misclassification of reads into other closely related taxa included in the DBs. Two profiles, BLASTN and Diamond+Megan, classified by the alignment-based software using large DBs suffered from the trade-off between recall and precision rates at the species level classification (BLASTN: higher recall, lower precision; Diamond+Megan: lower recall, higher precision). While profiles classified by the marker-based software, Metaphlan3, obtained relative high values in both recall and precision rates at its species level classification compared to other profiles in this study. For rest of the alignment-free software included in this study, all software reported both low precision and recall at their species-level classification. The number of distinct taxa that can be classified by each software are highly correlated with the number of reads each software can classify. Both distinct taxa identified, and number of reads classified are largely determined by the differences in DBs contents and the software's algorithms. Software with smaller DBs will not be able to identify species taxa that are not present in their DBs, thus requiring a smaller number of reads to reach a technical profiling threshold for the number of species they can classify. Software with large DBs can report species taxa that are not present in the sample due to failure of searching for the best match in a larger search space, shorter read lengths, or due to sequencing errors in both the query and the sequences contained in the DBs [72, 73]. The exceptions in our study are the profiles obtained using the Diamond+Megan and Metaphlan3, where the former one binned reads classified after profiling using MEGAN to increase the accuracy of each taxon's assignment [13, 35, 51] and the latter one could only identify the presence of taxa with marker sequences available in their customized DB [49]. Furthermore, with the simulated dataset, we have validated that the differences between taxonomy levels used by different DBs could also be a source of biases during microbial profiling [13, 74, 75], where the discrepancies in taxonomy levels between different DBs used for the classification could lead to inconsistencies in taxonomic profiles at higher taxonomic levels for microbial taxa profiling [76].

In general, with large differences in the distinct microbial taxa identified and the overall abundance profiles reported by different DBs and software, it is not surprising that microbial community characterization metrics will largely impact by the selection of software and DBs. For microbial community characterization within-sample, profiles of the index-based software have reported the largest divergences (Kaiju and Centrifuge) and reported significant differences in all metrics with most of the other profiles. In addition, metrics, such as Simpson indices, weigh more on the dominant species within each sample for microbial community characterization, are least impacted by the differences across profiles of the different software/DBs compared to metrics that weigh dominant and rare taxa more similarly (Shannon indices). For relationships between microbial communities determined from different profiles, the largest variations within the samples of a dataset could be characterized by profiles classified by all software/DBs. However, the statistical significance in evaluating biological variations between profiles is dissimilar when using profiles of different software/DBs, which could lead to inconsistent biological conclusions when different software/DBs are used for profiling.

Differentially abundant analysis is a frequently used statistical method used to determine the cause and outcome of infection or clinical treatments [77, 78]. In our study, only a small percentage of DA taxa identified using different profiles intersected even at the phylum levels and the intersection percentage decreased rapidly after class level, with most profiles reporting large numbers of distinct DA taxa when compared with other profiles. Due to high discrepancies in DA taxa reported at higher taxonomic levels, we suggest that direct-read shotgun metagenomics profiling approach should be taken with care when used for differential abundant analysis. Instead, profiling based on contigs mapping after assembling metagenome might be a better alternative for DA taxa identification.

The detection of *Leptospira* from the simulated samples has shown a high probability of false positive detections when using direct read metagenomics profiling for pathogen detection. However, false positive detections could be corrected by filtering out taxa with less than 1% in abundance. When identifying the presence of *Leptospira* in the rat samples, we found that profiles of most software could identify the presence of *Leptospira* from the samples that have reported *Leptospira* positive by traditional laboratory protocols. The profiles classified by Metaphlan3 were the least sensitive in *Leptospira* diagnostics, which failed to report the presence of *Leptospira* in a sample reported positive by all laboratory methods and by all the other profiles. Centrifuge, Kaiju, and Kraken2 with maxikraken2 DB (Kraken2_max) reported the presence of *Leptospira* in a sample reported *Leptospira* positive only by one out of three laboratory methods (PCR). However, Centrifuge and Kraken2_max, have also reported pathogenic *Leptospira* in a sample that has not been reported *Leptospira* positive by the traditional laboratory protocols, suggesting increased sensitivity for this method. For species-level diagnostics, the identity and number of *Leptospira* species reported by profiles classified by different software are largely diverse. However, filtering out taxa with less than 1% in abundance from each sample largely increased the consistency in species-level *Leptospira* detection across the different profiles. However, the filtering also decrease the sensitivity of the diagnostics compared to the laboratory methods.

In conclusion, the selection of software and DBs in direct-read shotgun metagenomics profiling can largely impact the profiling results as well as the microbial community characterization, differential abundant taxa, and pathogen detection. Algorithms used by the profiling software and DB contents are the major contributors for the differences between taxonomic profiles. In general, species-level microbial profiling and pathogen detection using shotgun metagenomics sequence reads are still challenging with the current profiling software due to sequencing error, DB contents, and short read length. Software with high recall rates (BLASTN) also suffers from low precision rates, and vice versa (Diamond+Megan). Metaphaln3 classifies profiles with a good balance between both metrics for bacteria classification. However, its profiles also suffer from low sensitivity when used for pathogen detection. Filtering out less abundant taxa, the inclusion of host genomes or genomes of interest in the DBs, or the use of long-read sequencing could be used to improve species-level accuracy [67, 69, 79–81]. Users must choose the appropriate software based on the goals of the study. For example, in pathogen detection studies, software with high sensitivity such (ex. Kaiju) or DBs with larger contents (Kraken2_max) may be selected, however, a follow-up laboratory validation is also important to further confirm the presence of the pathogen. When characterizing the microbial community, Kraken2 analysis with the use of the customized DBs for the inclusion of the host genomes along with the Bracken add-on can be a time and computationally efficient alternative for BLASTN and Diamond + Megan. To identify the most dominant species in a microbial ecosystem, the selection of the software will not largely impact the conclusion of the study.

The advancement in sequencing as well as in computational technologies allows modern-day biological research to move to a brand-new era. However, while benefiting from the power and convenience of technologies, we should always critically analyze and validate software outputs based on our prior knowledge. The inconsistencies found between the results of different metagenomic software showed that significant biological conclusions from metagenomic profiling analyses have the potential to be only the artifacts of the software' algorithms. We suggest researchers from different study fields to be aware of the possible error-prone conclusions made from metagenomics profiling analysis, and evaluate it, objectively comparing it to other traditional methods (e.g. PCR, culture, or serotyping).

## Supporting information

**S1 File.**
(DOCX)

**S1 Table. The complete profiling results obtained from 9 different taxnomic profiling software and four different Kraken2 DBs.**
(XLSX)

**S2 Table.** 1. Alpha indices (Observed, Shannon, and Simpson) obtained from profiles of all samples. 2. Pairwise comparisons for alpha indices obtained from profiles classified by different software+DBs combinations across samples. Paired Wilicoxon signed-rank test was used for statistical comparison. P-value was adjusted using the "holm" method.
(XLSX)

**S3 Table.** 1. Bray_curtis indices describing the between sample relationships were obtained pairwisely between samples using all profiles. 2. Pairwise comparisons for BC indices profiles obtained differet software + DBs' profiles. Statistical comparisons performed with paired wilicoxon signed rank test, p-value adjusted with "holm" method.
(XLSX)

**S4 Table.** 1. Differentially Abundant taxa (padj<0.05) identified from all the kidney samples vs. all the lung samples in the dataset using different profiles. 2. Differentially Abundant taxa (padj<0.05) identified from all the spleen samples vs. all the lung samples in the dataset using different profiles. 3. Differentially Abundant taxa (padj<0.05) identified from all the kidney samples vs. all the spleen samples in the dataset using different profiles.
(XLSX)

**S5 Table. Number of unique *Leptospira* species identified from rat kidney samples.** The number within each cell indicates the number of unique *Leptospira* species identified from each sample using the corresponding software in each row. The number inside the parenthesis of each cell indicates the number of pathogenic *Leptospira* species identified from the profiles classified by the corresponding software of each sample.
(XLSX)

## Author Contributions

**Conceptualization:** Sreekumari Rajeev, Liliana C. M. Salvador.

**Data curation:** Ruijie Xu.

**Formal analysis:** Ruijie Xu.

**Funding acquisition:** Sreekumari Rajeev, Liliana C. M. Salvador.

**Investigation:** Ruijie Xu.

**Methodology:** Ruijie Xu, Sreekumari Rajeev.

**Project administration:** Sreekumari Rajeev, Liliana C. M. Salvador.

**Resources:** Sreekumari Rajeev.

**Software:** Ruijie Xu.

**Supervision:** Sreekumari Rajeev, Liliana C. M. Salvador.

**Validation:** Ruijie Xu, Sreekumari Rajeev, Liliana C. M. Salvador.

**Visualization:** Ruijie Xu.

**Writing – original draft:** Ruijie Xu.

**Writing – review & editing:** Sreekumari Rajeev, Liliana C. M. Salvador.

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
