## [Decision Letter · Decision Letter 0]

6 Dec 2022

PONE-D-22-23909The selection of software and database for metagenomics sequence analysis impacts the outcome of microbial profiling and pathogen detectionPLOS ONE

Dear Dr. Salvador,

Thank you for submitting your manuscript to PLOS ONE. After careful consideration, we feel that it has merit but does not fully meet PLOS ONE’s publication criteria as it currently stands. Therefore, we invite you to submit a revised version of the manuscript that addresses the points raised during the review process.

 Please see the detailed comments below for more information.

We look forward to receiving your revised manuscript.

Kind regards,

Brian B. Oakley, PhD

Academic Editor

PLOS ONE

Journal Requirements:

2. "PLOS requires an ORCID iD for the corresponding author in Editorial Manager on papers submitted after December 6th, 2016. Please ensure that you have an ORCID iD and that it is validated in Editorial Manager. To do this, go to ‘Update my Information’ (in the upper left-hand corner of the main menu), and click on the Fetch/Validate link next to the ORCID field. This will take you to the ORCID site and allow you to create a new iD or authenticate a pre-existing iD in Editorial Manager. Please see the following video for instructions on linking an ORCID iD to your Editorial Manager account: " ext-link-type="uri" xlink:type="simple">https://www.youtube.com/watch?v=_xcclfuvtxQ"

"The sequence analysis work was supported by the National Science Foundation under Grant No. DGE-1545433 to R.X. and startup funds to L.C.M.S. from the University of Georgia Office of Research. The sample collection, sequencing and analysis was done during S.R.’s tenure at the Ross University School of Veterinary Medicine, Saint Kitts and it was supported by internal grants from the Center for One Health and Tropical Medicine. We also would like to thank Dr. Kanae Shiokawa for her help with collection and processing of rat specimens."

Additional Editor Comments (if provided):

In light of the difficulty that has been experienced finding qualified reviewers for this manuscript and to avoid further delay, I am providing some comments in addition to those provided by Reviewer #1. As the points raised by the reviewer are addressed, please also address these additional issues.

The first point follows a comment by Reviewer #1 regarding the use a control sample or one with some sort of an internal standard to help provide some absolute benchmarks in the comparison of these software tools. Since the results varied dramatically for multiple parameters, this begs the important question of which platform is best for whatever criteria may be of interest to users. In the Introduction section, some previous work is cited regarding synthetic datasets and so clearly some of this work has been done, but it is still an important requirement to have appropriate positive and negative controls in any dataset. This could be done in various ways - e.g. construction of an in silico dataset, qPCR of Leptospira from the biological samples used here, computational 'spiking in' of known quantities of Leptospira variant sequences, etc. Please address this important issue that will greatly help readers make meaningful choices as to the value of the software packages and databases compared. 

Second, some more care and strategic decisions should be made regarding the data presented. Many figures and tables (some as supplementary) are presented, but the points that are attempting to be made are not always clear. One minor example is in Table 1 where incomplete information is provided about the computational resources required. This information does not always make sense - for example, how were 0 Mb of 'resources' used for Bracken?

Finally, one additional analysis that might be interesting would be a beta-diversity analysis such as a PCoA of results obtained, testing for significant clustering according to approach (e.g. alignment vs k-mer based).

As noted in the manuscript, the increasing use of metagenomics makes a manuscript such as this potentially very useful to the field. I feel the changes above should significantly strengthen the work.

Reviewers' comments:

Reviewer's Responses to Questions

**Comments to the Author**

1. Is the manuscript technically sound, and do the data support the conclusions?

Reviewer #1: Yes

2. Has the statistical analysis been performed appropriately and rigorously? 

Reviewer #1: Yes

3. Have the authors made all data underlying the findings in their manuscript fully available?

Reviewer #1: No

4. Is the manuscript presented in an intelligible fashion and written in standard English?

Reviewer #1: Yes

5. Review Comments to the Author

Reviewer #1: The manuscript addresses a relevant piece of information on several metagenomics profiling tools' performance and outcome discrepancies. Its content is undoubtedly attractive to researchers in the field, raising awareness and caution on software selection and results interpretation. It is both technically and scientifically sound, however lacks expatiating on important technical features of a full metagenomics workflow, which I better explain through the comments below.

1. Even clearly understanding that taxonomic profiling and taxa abundance statistical analyses were the most important aspects of the software comparison's scope of this manuscript, I feel like it lacks information on assembly steps (assembly, binning, bins' reassembly) of WGS metagenomics, as well as ESVs clustering for 16S-NGS. I encourage authors to open a paragraph on either Methods (if they have used any metagenomics assembler for the preparation of this manuscript) or Results section expatiating on the importance of both assembly/binning (metagenomics) and ESV generation (16S) steps within a metagenomics/microbiomics pipeline.

2. Was there any "true" dataset used to rely on for Precision calculations on Fig.3? This must be briefly explained it in Fig.3 caption. Who are the true positive taxa and where do they come from?

3. line 366: Replace "prevision" by "precision".

4. line 449: Although aware that BC acronym was first introduced on the Methods section, I'd rather have it written out again on the first time it appears in the Results. So, Bray-Curtis (BC) on line 449, as well. Many readers may ignore Methods section if they're not planning to do a similar work.

5. Fig 7a is an "UpSet Venn Diagram". Please introduce it as such on its caption.

6. Considering what is displayed in Fig 7b (summed to my #1 comment in this review), it came to my attention that a different, and potentially better, outcome might have been achieved if each sample had their own generated bins or MAGs (metagenome-assembled genomes) to be compared against each other through a pseudoalignment approach (e.g. raw reads in sample_A pseudoaligned against MAGs from sample_B and raw reads in sample_B pseudoaligned against MAGs from sample_A). Authors must consider expatiating on that as well for this particular result interpretation.

7. Ok, just saw lines 675-677 from Discussion where assembly and binning strategies were finally mentioned (although quite briefly). It doesn't exclude the need of addressing my comments #1 and #6 above, which will reinforce what is currently briefly mentioned.

8. line 680: Replace "is" by "are".

9. Please make sure figures will have a better resolution for publication.

10. Raw sequencing data must be deposited in a public repository (e.g. SRA from NCBI), and SRR accession numbers provided in the manuscript (or a NCBI BioProject ID that will point to the SRRs).

6. PLOS authors have the option to publish the peer review history of their article (what does this mean?). If published, this will include your full peer review and any attached files.

Reviewer #1: No

---

## [Author Response · Author response to Decision Letter 0]

1 Mar 2023

Responses to Editor and Reviewer’s Comments

PONE-D-22-23909

The selection of software and database for metagenomics sequence analysis impacts the outcome of microbial profiling and pathogen detection

PLOS ONE

Additional Editor Comments (if provided):

In light of the difficulty that has been experienced finding qualified reviewers for this manuscript and to avoid further delay, I am providing some comments in addition to those provided by Reviewer #1. As the points raised by the reviewer are addressed, please also address these additional issues.

The first point follows a comment by Reviewer #1 regarding the use a control sample or one with some sort of an internal standard to help provide some absolute benchmarks in the comparison of these software tools. Since the results varied dramatically for multiple parameters, this begs the important question of which platform is best for whatever criteria may be of interest to users. In the Introduction section, some previous work is cited regarding synthetic datasets and so clearly some of this work has been done, but it is still an important requirement to have appropriate positive and negative controls in any dataset. This could be done in various ways - e.g. construction of an in silico dataset, qPCR of Leptospira from the biological samples used here, computational 'spiking in' of known quantities of Leptospira variant sequences, etc. Please address this important issue that will greatly help readers make meaningful choices as to the value of the software packages and databases compared. 

We thank the editor for the time and effort taken to help us improve our manuscript.

To address this suggestion, we added an in silico dataset to benchmark the different software and databases (DBs) used in this study. Specifically, three simulated samples of mice gut microbiome with known taxonomic profiles were added to the manuscript to use as a control for the comparison between profiles classified using different software and DBs. Using these simulated samples, we obtained recall and precision rates for all the profiles classified with the different software and DBs. This addition also allowed us to assess the accuracy of Leptospira detection using the simulated samples. For the Leptospira detection section, laboratory diagnostics results (previously published in Rajeev et al. 2020) were also added to the manuscript, serving as another benchmark for Leptospira detection.

Rajeev, S., Shiokawa, K., Llanes, A., Rajeev, M., Restrepo, C. M., Chin, R., Cedeño, E., Ellis, E. (2020). Detection and Characterization of Leptospira Infection and Exposure in Rats on the Caribbean Island of Saint Kitts. Animals, 10(2), 350. https://doi.org/10.3390/ani10020350

Lines 115-123: “The simulated mice gut microbiome dataset was obtained from a metagenomics software benchmarking project, the Critical Assessment of Metagenome Interpretation (CAMI) initiative (38), available at (https://doi.org/10.4126/FRL01-006421672). This dataset includes 64 simulated mice gut microbiome samples from 12 different mice with samples both simulated as Illumina (pair-end, 150bp) and PacBio reads (~3000 bps/read) using NCBI’s RefSeq genomes (39). Only the first three simulated Illumina reads samples (~5GB per sample) were used in this study to produce a standardized evaluation (precision and recall) for the profiling software and DBs included in this study (2017.12.29_11.37.26_sample_0 (Sim.0), 2017.12.29_11.37.26_sample_1 (Sim.1), and 2017.12.29_11.37.26_sample_2 (Sim.2)).”

We added a figure (new Figure 2) and several sentences throughout the manuscript presenting and discussing the results associated to the simulated datasets. 

Second, some more care and strategic decisions should be made regarding the data presented. Many figures and tables (some as supplementary) are presented, but the points that are attempting to be made are not always clear. One minor example is in Table 1 where incomplete information is provided about the computational resources required. This information does not always make sense - for example, how were 0 Mb of 'resources' used for Bracken?

We thank the editor for alerting us to this fact. We moved Figure 3 into the supplementary materials (currently Figure S3), removed Figure 4 from the previous version of the manuscript, and added Figures 2 and 4 in the current version of the manuscript (previous Figure2 - current Figure3) to make the manuscript more concise and informative. We also edited Table 1 for improved clarity. Specifically, the column “Resources” was renamed “memory usage”. For the 0 memory usage of Bracken during profiling correction, this was reconfirmed by the resource usage reported by the University of Georgia’s high performance computing cluster system. 

Finally, one additional analysis that might be interesting would be a beta-diversity analysis such as a PCoA of results obtained, testing for significant clustering according to approach (e.g. alignment vs k-mer based).

A PCoA plot showing beta diversities between the profiles classified by the different software and DBs is now presented in Figure 3 of the manuscript. We aggregated the profiles of the different rat samples together to produce one single profile for each software/DBs and visualized the relationships between profiles using PCoA. Furthermore, significant clustering based on the algorithms of each software was tested using the PERMANOVA test.

Lines 407-408: “Fig 4. PCoA plot visualizing relationships between profiles classified by different software. The profile of each software is colored based on the type of algorithm it was developed with.”

As noted in the manuscript, the increasing use of metagenomics makes a manuscript such as this potentially very useful to the field. I feel the changes above should significantly strengthen the work.

We thank the editor for the endorsement of our work.

Reviewers' comments:

Reviewer's Responses to Questions

Comments to the Author

1. Is the manuscript technically sound, and do the data support the conclusions?

Reviewer #1: Yes

2. Has the statistical analysis been performed appropriately and rigorously?

Reviewer #1: Yes

3. Have the authors made all data underlying the findings in their manuscript fully available?

Reviewer #1: No

4. Is the manuscript presented in an intelligible fashion and written in standard English?

Reviewer #1: Yes

5. Review Comments to the Author

Reviewer #1: The manuscript addresses a relevant piece of information on several metagenomics profiling tools' performance and outcome discrepancies. Its content is undoubtedly attractive to researchers in the field, raising awareness and caution on software selection and results interpretation. It is both technically and scientifically sound, however lacks expatiating on important technical features of a full metagenomics workflow, which I better explain through the comments below.

1. Even clearly understanding that taxonomic profiling and taxa abundance statistical analyses were the most important aspects of the software comparison's scope of this manuscript, I feel like it lacks information on assembly steps (assembly, binning, bins' reassembly) of WGS metagenomics, as well as ESVs clustering for 16S-NGS. I encourage authors to open a paragraph on either Methods (if they have used any metagenomics assembler for the preparation of this manuscript) or Results section expatiating on the importance of both assembly/binning (metagenomics) and ESV generation (16S) steps within a metagenomics/microbiomics pipeline.

We thank the reviewer for the suggestion. We agree with the reviewer that assembly, binning, and NGS 16S are very important aspects in the field of metagenomics. However, the goal of this manuscript is to evaluate the performances of different direct-read shotgun metagenomics profiling software, and their biases introduced during community characterization and pathogen detection in the downstream analyses. To provide a complete introduction to the type of metagenomics profiling methods available, we added a paragraph in the introduction section discussing the use of direct-read profiling and assembly/binning-based metagenomics analyses. Following the reviewer’s suggestion, we also attempted to assemble and bin the contigs of our datasets, however, due to the large percentage of host contamination in our data, only one-third of the samples could obtain bins after the assembly procedure and these were less than two bins each. We also discussed this finding in the Discussion section.

Lines 67-76: “The microbial classification of shotgun metagenomics sequencing could also be divided into two primary categories: direct read profiling and assembly-based profiling (13), where software developed under each category was developed to answer different research questions. Direct read profiling software aim to quantitively characterize the microbial communities of the collected samples (e.g. species diversity and richness) (21), distinguish the presence of disease-causing pathogens from their non-pathogenic close relatives (22), and identify new microbial organisms (23). While assembly-based classification software mainly aims to qualitatively characterize the complete genomes of uncultivated microbial organisms (24) or understand the metabolic functions of the microbial community through gene or metabolic pathway characterization (using metagenome assembly and contig binning (25,26)).”

2. Was there any "true" dataset used to rely on for Precision calculations on Fig.3? This must be briefly explained it in Fig.3 caption. Who are the true positive taxa and where do they come from?

There was no true dataset used for the “relative precision” rate calculated in Figure 3 of the old version of the manuscript. The relative precision rate was calculated between each pair of profiles of the different software/DBs. The detailed formula and explanation of the “relative precision” rate were defined in the Materials and Methods section. However, we realize that this metric may lead to confusion in the manuscript and to improve clarity, the old Figure 3 was moved to the Supplementary Materials (Supplementary Figure 3), and the new Figure 3 in the revised manuscript was replaced by the analysis describing the relationships between profiles of the different software/DBs using a PCoA plot, following the suggestion of the editor.

Lines 234-241: “Distinct species taxa identified from all profiles of the rat tissue samples were compared in a pairwise fashion, where we defined a comparative metric, relative precision rate, to describe the differences and similarities between the distinct microbial taxa identified between two profiles classified using two different software/DBs included in this study. Relative precision rate is defined as the percentage of intersection in taxa identified from two different profiles included in a comparison (A vs. B) relative to the total number of microbial taxa identified by the profile A within this comparison (|AB||A|). Relative precision analysis was performed using a custom R script.

We also assessed the between profiles relationship of different software/DBs using the Bray-Curtis (BC) indices, where we aggregated the number of reads classified under each microbial taxon identified from all the rat samples together to obtain a single taxonomic profile for each software/DB. The relationships between these aggregated profiles were visualized with a principal coordinate analysis (PCoA) plot using the “phyloseq” package in R (52).”

3. Line 366: Replace "prevision" by "precision".

The typo was part of the figure caption for Figure 3 in the previous version of the manuscript. This figure was moved to the Supplementary Material (Figure S3) and the typo was corrected.

Supplementary Figures Lines 46-50: “Fig S3. Relative precision rates for the distinct microbial taxa identified at the species level between profiles identified by the different software and DBs. The boxplots on each side represent the relative precision rates across samples being compared (A vs. B), where relative precision rates of profiles A and B are presented on the left- and right-hand sides, respectively.”

4. Line 449: Although aware that BC acronym was first introduced on the Methods section, I'd rather have it written out again on the first time it appears in the Results. So, Bray-Curtis (BC) on line 449, as well. Many readers may ignore Methods section if they're not planning to do a similar work.

Bray-Curtis (BC) indices ware redefined in the Results section.

Lines 438-440: “We characterized the between sample relationship with the Bray-Curtis (BC) dissimilarity indices (Table S3.1) and visualized the relationships across samples using the principal coordinate analyses (PCoA) plots.”

5. Fig 7a is an "UpSet Venn Diagram". Please introduce it as such on its caption.

The “upSet Venn Diagram” was added to the caption of Figure 7.

Lines 475-476: “Fig 7. UpSet Venn Diagram showing the intersection in DA taxa identified between the kidney and lung samples by different software and DB profiles. ”

6. Considering what is displayed in Fig 7b (summed to my #1 comment in this review), it came to my attention that a different, and potentially better, outcome might have been achieved if each sample had their own generated bins or MAGs (metagenome-assembled genomes) to be compared against each other through a pseudoalignment approach (e.g. raw reads in sample_A pseudoaligned against MAGs from sample_B and raw reads in sample_B pseudoaligned against MAGs from sample_A). Authors must consider expatiating on that as well for this particular result interpretation.

We thank the reviewer for the suggestion, and we agree with the reviewer that MAGs or bins might be able to achieve higher consistency in the differentially abundant taxa identified using profiles produced by the different software and DBs. In the discussion of the current manuscript, we suggested that direct-read profiling results may have to be taken with care when used for the identification of the differentially abundant taxa due to high discrepancies in DA taxa reported and that MAGs might be a better option. However, we didn’t directly perform this analysis in the current version of the manuscript because the aim of our study was solely to focus on the biases introduced by the selection of different direct-read profiling software and databases on the taxonomic profiles.

Lines 621-625: “Due to high discrepancies in DA taxa reported at higher taxonomic levels, we suggest that direct-read shotgun metagenomics profiling approach should be taken with care when used for differential abundant analysis. Instead, profiling based on contigs mapping after assembling metagenome might be a better alternative for DA taxa identification.”

7. Ok, just saw lines 675-677 from Discussion where assembly and binning strategies were finally mentioned (although quite briefly). It doesn't exclude the need of addressing my comments #1 and #6 above, which will reinforce what is currently briefly mentioned.

We thank the reviewer for the suggestions. We added detailed information about the use of assembled-based profiling and direct read profiling in the introduction section. We also attempted to assemble and bin the reads in our datasets, however, due to host DNA contamination, only a few contigs from each sample were assembled, and only 1/3 of the samples were able to cluster their assembled contigs into bins (less than 2 bins from each sample). We discussed this analysis in the discussion and emphasized the impact of host DNA contamination as one of the major factors in biasing metagenomics analysis.

Lines 67-76: “The microbial classification of shotgun metagenomics sequencing could also be divided into two primary categories: direct read profiling and assembly-based profiling (13), where software developed under each category was developed to answer different research questions. Direct read profiling software aim to quantitively characterize the microbial communities of the collected samples (e.g. species diversity and richness) (21), distinguish the presence of disease-causing pathogens from their non-pathogenic close relatives (22), and identify new microbial organisms (23). While assembly-based classification software mainly aims to qualitatively characterize the complete genomes of uncultivated microbial organisms (24) or understand the metabolic functions of the microbial community through gene or metabolic pathway characterization (using metagenome assembly and contig binning (25,26)).”

Lines 553-560: “These host DNA contaminations in metagenomics sequenced samples will not only impact the accuracy of quantitative characterization for the microbial communities, but also will prevent the potential of performing qualitative analyses using shotgun metagenomics sequenced data. For example, to perform functional analysis using the rat tissue samples collected in this study requires the assembling and binning of the sequenced reads beforehand, however, only 4 out of 12 tissue samples were able to obtain contig bins after metagenome assembly+binning, with less than 2 bins clustered from each sample (data not shown).”

8. Line 680: Replace "is" by "are".

This sentence has been deleted from the current version of the manuscript.

9. Please make sure figures will have a better resolution for publication.

All figures are at least 500 dpi in TIFF format.

10. Raw sequencing data must be deposited in a public repository (e.g. SRA from NCBI), and SRR accession numbers provided in the manuscript (or a NCBI BioProject ID that will point to the SRRs).

All data used in this study has been uploaded to NCBI and are publicly available. Information about data availability can be find at the Data Summary section of the manuscript.

Lines 679-684: “The simulated mice gut microbiome dataset was obtained from a metagenomics software benchmarking project, the Critical Assessment of Metagenome Interpretation (CAMI) initiative (38), available at (https://doi.org/10.4126/FRL01-006421672).”

 “The raw sequence files (FASTQ) were submitted to the NCBI Sequence Read Archive under the Bioproject accession number: PRJNA717669. The individual samples can be accessed under the following Biosample accession numbers: SAMN18507082 - SAMN18507091.”

---

## [Editor Report · Decision Letter 1]

22 Mar 2023

The selection of software and database for metagenomics sequence analysis impacts the outcome of microbial profiling and pathogen detection

PONE-D-22-23909R1

Dear Dr. Salvador,

We’re pleased to inform you that your manuscript has been judged scientifically suitable for publication and will be formally accepted for publication once it meets all outstanding technical requirements.

Kind regards,

Brian B. Oakley, PhD

Academic Editor

PLOS ONE

Additional Editor Comments (optional):

Thank you for the revisions and responses to the previous comments.  After careful review, all of the points that were raised have been adequately addressed and the manuscript has been substantially improved.
---

## [Editor Report · Acceptance letter]

29 Mar 2023

PONE-D-22-23909R1 

The selection of software and database for metagenomics sequence analysis impacts the outcome of microbial profiling and pathogen detection 

Dear Dr. Salvador:

I'm pleased to inform you that your manuscript has been deemed suitable for publication in PLOS ONE. Congratulations! Your manuscript is now with our production department. 

Kind regards, 

on behalf of

Dr. Brian B. Oakley 

Academic Editor

PLOS ONE